# Pyrolysis Behaviors and Residue Properties of Iron-Rich Rolling Sludge from Steel Smelting

**DOI:** 10.3390/ijerph19042152

**Published:** 2022-02-14

**Authors:** Hengdi Ye, Qian Li, Hongdi Yu, Li Xiang, Jinchao Wei, Fawei Lin

**Affiliations:** 1National Engineering Research Center of Sintering and Pelletizing Equipment System, Zhongye Changtian International Engineering Co., Ltd., Changsha 410205, China; yehengdi_cie@163.com (H.Y.); weijinchao_cie@163.com (J.W.); 2School of Engineering, GongQing Institute of Science and Technology, Jiujiang 332020, China; 3Tianjin Key Lab of Biomass/Wastes Utilization, School of Environmental Science and Engineering, Tianjin University, Tianjin 300072, China; yuhongdi@tju.edu.cn (H.Y.); xiangli97@tju.edu.cn (L.X.)

**Keywords:** iron-rich rolling sludge, pyrolysis, solid residues, resource recovery

## Abstract

Iron-rich rolling sludge (FeRS) represents a kind of typical solid waste produced in the iron and steel industry, containing a certain amount of oil and large amounts of iron-dominant minerals. Pyrolysis under anaerobic environment can effectively eliminate organics at high temperatures without oxidation of Fe. This paper firstly investigated comprehensively the pyrolysis characteristics of FeRS. The degradation of organics in FeRS mainly occurred before 400 °C. The activation energy for pyrolysis of FeRS was extremely low, ca. 5.44 kJ/mol. The effects of pyrolytic temperature, atmosphere, heating rate, and stirring on pyrolysis characteristics were conducted. Commonly, the yield of solid residues maintained around 85 wt.%, with approximately 13 wt.% oil and 2 wt.% gas. Due to the low yield of oil and gas, their further utilization remains difficult despite CO_2_ introduction which could upgrade their quality. The solid residues after pyrolysis exhibited porous properties with co-existence of micropores and mesopores. Combined with the high content of zero-valent iron, magnetic property, hydrophobic characteristic, and low density, the solid residues could be further utilized for water pollution control and soil remediation. Moreover, the solid residues were suitable for sintering to recover valuable iron resources. However, the solid residues also contained certain heavy metals, such as Cd, Cr, Cu, Ni, Pb, and Zn, which might cause secondary pollution during their utilization. In particular, the toxic Cr possessed high content, which should be treated with detoxification and removal. This paper provides fundamental information for pyrolysis of FeRS and utilization of solid residues.

## 1. Introduction

The steel and iron industry as an important manufacturing category are essential to China’s economic development [1,2]. As estimated, the world’s production of crude steel was 1808 million tons in 2018, and China’s total output shares rose up to 51.32% [3]. At a conversion rate of 0.35–0.45%, a total of 37.1–47.1 million tons of iron-rich rolling sludge (FeRS) was produced in 2018 in China [4]. FeRS is a kind of industrial waste with dense slurry comprising of lubricating oil, iron finer, and other contaminants, which are generated from the steel rolling link [5,6,7]. The scrubbing solution during washing equipment enters the sludge tank and precipitates to form sludge, representing the primary source of FeRS [8]. The main components of FeRS are Fe, Fe_2_O_3_, FeO, and trace elements, such as Ca, Si, Mg, Na, Zn, and Pb [9]. Moreover, FeRS possesses a certain content oil, therefore the fluidity of FeRS is directly proportional to viscosity and oil content [10]. Furthermore, the accumulation and storage of FeRS not only occupies vast land resources, but also seriously threaten human health and the ecological environment due to its hazardous characteristics [11,12]. However, the FeRS also contains large amounts of iron resources after removal of harmful substances, e.g., heavy metals and petroleum hydrocarbons [13]. Organized collection of FeRS and developed applicable methods for harmless treatment and resource recycling can not only recover valuable iron resources with economic benefits, but also reduce pollution thereby obtaining an environment social benefit.

At present, the treatment methods of FeRS can be divided into two categories, i.e., physical and chemical methods. Physical methods include landfill [14,15], solvent extraction [16], microwave solidification [17], and hot water washing methods. Chemical methods include coking [18], dry incineration [19], thermal desorption and pyrolysis methods [10]. Moreover, a coupling method of quenching and tempering dehydration is also utilized [20,21]. The iron content present in FeRS is about 50–60%, possessing a high iron resource recovery value and potential in the steel process [22]. Landfill remains infeasible due to site contamination and waste of iron resources. Chemical extraction with high costs and low efficiency cannot be widely applied [15]. Incineration causes secondary pollutant emissions of NO_x_, SO_x_, and heavy metals [10,23,24]. Comparatively, pyrolysis treatment under anaerobic environment can effectively eliminate the organics in FeRS with recovery of oil content. The pyrolysis gas can still be mixed into the sintering surface for gas-phase heating. Hence, the effective recovery of FeRS can be realized, providing an efficient resource recovery and disposal method. Above all, pyrolysis bears excellent application prospects in the future.

FeRS as the object of exploring, represents a typical solid waste from the whole process of iron and steel plants. The typical characteristics of FeRS are identified from the three perspectives of energy utilization, resource utilization, and toxic substance properties. However, the pyrolysis characteristics of FeRS remain unknown. This paper firstly obtained the basic characteristic data of FeRS based on industrial analysis, element analysis, and heavy metal analysis of iron- containing sludge. Moreover, TG–FTIR–MS (Thermogravimetric–Fourier Transform Infrared Spectrometer–Mass Spectrometry) was employed to monitor the outgassing and releasing law of FeRS pyrolysis process online. The product’s distribution and quality with varied pyrolytic parameters were explored. Furthermore, the physicochemical properties of solid residue as well as internal heavy metals were analyzed to evaluate further utilization routes.

## 2. Material and Experimental Section

### 2.1. Pyrolytic Apparatus

FeRS sample was provided by Zhongye Changtian International Engineering Co., Ltd. (Hunan province, China). To collect the pyrolytic products during FeRS pyrolysis, a vertical fixed-bed reactor was designed [11,12]. In each experiment, ~60 g sample dried at 105 °C was firstly loaded at the bottom of a left stainless tube, and then underwent a temperature ramping process. The flow rate of background gas (N_2_ or CO_2_) was fixed at 100 mL/min. The gas entered from the bottom of the left tube, subsequently returned to rise and exited through the bottom of right tube, which was denoted by yellow dotted lines. Next, the outgoing gas was condensed by −10 °C ethanol to collect pyrolytic oil by a round-bottom flask, while the non-condensing gas was collected by a gas bag for further tests. Moreover, a stirrer controlled by an external motor was conducted to improve the transfer of heat and mass during pyrolysis. In this study, a series of single-factor variable experimental researches was conducted on the effect of ending temperature, heating rate, stirring rate, and reaction atmosphere. Table 1 lists the reaction conditions represented by each designation.

### 2.2. Pyrolytic Products

After pyrolysis, the yields of oil and solid residue were weighed, and the yield of gas was calculated based on mass balance. According to the national standard GB/T 212-2008, the content of moisture, ash, volatile, and fixed carbon of FeRS were obtained by industrial analysis. Specifically, the content of moisture was determined by placing 1 g sample in an oven and dried at 105 °C to a constant weight. Ash content was obtained by weighing the remaining residue after the sample burned at 815 °C for 40 min in a muffle furnace. The volatiles content marked the mass reduction of the sample after burning at 900 °C for 7 min and fixed carbon content was obtained based on mass conservation. Elemental analysis (C/H/O/N/S) was tested by the combustion method, using an elemental analyzer (Elementat Vario ELIII) with a detection limitation of 1 ppm. The heating value of FeRS was measured by oxygen bomb combustion and calorimeter.

To evaluate the distribution of heavy metals in solid residues, HNO_3_–HF–HClO_4_ digestion was employed for the pretreatment and the obtained solution was monitored by inductive coupled plasma mass spectrometry (Agilent 7500A). The relative contents of components in pyrolytic oil dissolved by dichloromethane were quantitively analyzed by gas chromatography-mass spectrometry (GCMS, QP2010 SE). RTX-5MS 5% diphenyl-95% dimethyl polysiloxane (0.25 mm × 30 m × 0.25 µm) was used as GC column. The initial temperature program was as follows: 50 °C (held for 3 min) to 300 °C (10 °C/min, held for 5 min). The carrier gas was He and constant voltage of 136.8 kPa was conducted to control flow rate. The temperature of the vaporization chamber was set at 260 °C and the split ratio was 50:1. MS detector used electron ionization (EI) mode. The temperatures of ion source and interface were 200 °C and 280 °C, respectively. Data were collected under full scanning mode with m/z ranging from 35 to 500 and then identified according to NIST (National Institute of Standard Technology) database. The pyrolytic gas collected in gas bag was analyzed by gas chromatograph (GC, Agilent 7890A). The mineral compositions of FeRS were determined by X-ray fluorescence spectrometry (XRF, PANalytical AXIO MAX), and the detection limitation was 1 ppm. The specific surface areas and pore structures of solid residues were investigated by nitrogen adsorption instrument (BELSORP-max). The Brunauer–Emmett–Teller (BET) model was used to analyze surface area. Subsequently, the quench solid state density functional theory (QSDFT) was used to calculate pore distribution. The specific surface areas and pore volumes of micropores were calculated according to the Horvath–Kawazoe (HK) model. The total pore volume was determined by the amount of N_2_ adsorbed at a relative pressure of 0.99. The morphologies of solid residues were detected on a Merlin scanning electron microscope (SEM). The X-ray diffraction (XRD) patterns were obtained on a diffractometer (X’ Pert Pro XRD) using Cu Kα radiation (λ = 1.54056 Å, 10° min^−1^ from 10 to 90°). The XRD data were imported into MDI Jade 6 software for further analysis. X-ray photoelectron spectra (XPS) were recorded on an ESCALAB 250Xi analyzer using a monochromatic Al Kα source, and the binding energies were referenced to C 1s peak at 284.6 eV.

### 2.3. Thermochemistry of FeRS

The weight loss curves during pyrolysis were detected by thermal gravimetric analyzer (TG, Netzsch STA 449F3 Jupiter). In each experiment, ~15 mg FeRS was loaded into a TG crucible. The temperature gradient caused by agglomeration could be ignored due to the small amount of samples. The heating procedures were set from 50 to 900 °C with a varied ramping rate of 10/20/30/40/50 °C/min. Argon was used as the background gas and the flow rate was maintained at 50 mL/min. Further, the basic equation of pyrolytic kinetics was obtained according to Arrhenius law, as shown in Equations (1) and (2).
(1)$$\frac{d\alpha }{dT}=\frac{A}{\beta }f\left(\alpha \right)\exp\left(\frac{-E_{a}}{RT}\right)$$
(2)$$f\left(\alpha \right)={\left(1-\alpha \right)}^{n}$$
where, *T* is pyrolysis temperature, *K*; *A* is the pre-exponential factor, *K*^−1^; *Ea* is the apparent activation energy, kJ/mol; *β* is the heating rate, K/min; *α* is conversion rate; *F(α)* is the pyrolysis reaction mechanism model, and *n* is the reaction order. *R* is the ideal gas constant 8.314 J·mol^−1^ k^−1^.

Assuming the FeRS pyrolysis followed first-order reaction (*n* = 1), the above equation could be simplified as:(3)$$\frac{d\alpha }{dT}=\frac{A}{\beta }f\left(\alpha \right)\exp\left(\frac{-E_{a}}{RT}\right)$$

After derivation of Doyle integral and Hancock empirical formula, the simplification could be acquired:(4)$$ln\left[-ln\left(1-\alpha \right)\right]=-\frac{E_{a}}{RT}+\left(ln\frac{AE_{a}}{R\beta }-5.33\right)$$

As can be seen, *ln*[−*ln*(1 − *α*)] has a linear relationship with *T*^−1^. Therefore, preexponential factor *A* and apparent activation energy *Ea* of each stage can be calculated according to intercept and slope, respectively.

## 3. Results and Discussion

### 3.1. Physicochemical Characteristics

The basic characteristics of FeRS were carried out and evaluated. Table 2 and Table 3 tabulate the element analysis, heating value, and mineral compositions of FeRS. FeRS possessed lower contents of C, H, O, N, and S, ca. 11.05%, 1.66%, 7.07%, and 0.13%. These compositions mainly originated from lubricating oil. Furthermore, the high heating value (HHV) and low heating value (LHV) of FeRS were also tested, ca. 10.13 and 9.80 kJ/g. The oxidation of Fe also contributed to the calorific value [5]. Regarding the XRF results shown in Table 3, FeRS included 10 minerals, including Na_2_O, Al_2_O_3_, SiO_2_, P_2_O_5_, CaO, Cr_2_O_3_, MnO, Fe_2_O_3_, NiO, CuO, ZnO, and MoO_3_. Furthermore, Fe_2_O_3_ occupied the highest proportion, ca. 57.15 wt.%. Al_2_O_3_, SiO_2_, and Cr_2_O_3_ also exceeded 1.00 wt.%, ca. 2.17 wt.%, 1.48 wt.%, and 1.14 wt.%, respectively. Above all, FeRS possesses a low content of organics, high content of metal elements, and a higher calorific value. The disposal principle of FeRS should focus on the elimination of organics and final utilization of residues.

### 3.2. TG Analysis of FeRS

The TG (Thermogravimetric Analysis) and DTG (Differential Thermogravimetric Analysis) curves of FeRS at different heating rates are shown in Figure 1. The TG curves of FeRS at different heating rates exhibited that weight loss mainly occurred between 100–380 °C. The moisture and organics of FeRS volatized within this stage. Slight weight loss could also be detected between 380–600 °C but possessed a weak weight-losing rate as the shoulder appeared in DTG curves. Heavy organics and secondary decomposition of coke were released in this interval. Following this, transformation of minerals contributed to the final weight loss above 600 °C. This phenomenon was shown in our previous study which related to the weight loss [7,25,26]. Finally, the total weight loss of FeRS ranged between 13 and 20 wt.%, which reflected the content of organics. From observing the DTG curves, the weight loss rate of FeRS increased continuously with the elevation of heating rate. Additionally, the hump at approximately 250 °C for low heating rates tended to disappear when the heating rate reached 50 °C/min. This might be due to the shortening time of organic components to reach the corresponding releasing temperature, which was consistent with other studies [27]. According to the TG and DTG curves, the main pyrolysis stage (100–600 °C) of FeRS was calculated by differentiational method (Figure 2 and Table 4). The correlation coefficients between fitted straight line and the actual calculated data were all above 0.92, indicating the first-order reaction was appropriate to the pyrolysis process of FeRS. The activation energy of FeRS varied between 4.26 and 5.44 kJ/mol in different heating rates, which was much lower than oily sludge. Comparatively, the activation energy of oily sludge reached up to 12.08 kJ/mol under the same heat treatment in previous studies [7]. Therefore, the activation energy exhibited a weak relationship with heating rate. The low oil content and weak emulsification contributed to the lower activation energy of FeRS [25].

### 3.3. Effect of Temperature and Atmosphere

#### 3.3.1. Products Distribution

Figure 3 displays the correlation between pyrolysis temperature/atmosphere and pyrolytic products distribution. As expected, the yield of solid residues occupied dominant position, ca. 79.7–87.4 wt.%, consistent with total weight loss from TG results. The oil yield ranged between 12.0–17.2 wt.% for all conditions, while the presence of pyrolytic gas was negligible. With the elevation of pyrolysis temperature, the oil yield exhibited a slight difference due to the almost complete release of organics before 400 °C. Moreover, different behaviors between the yield of oil and gas could also be observed, which was attributed to the transformation between oil and gas. As per the TG and furnace results, pyrolysis at 500 °C was sufficient to achieve complete elimination of organics in FeRS. Therefore, further investigations for pyrolysis of FeRS were conducted at a fixed pyrolytic temperature, ca. 500 °C. CO_2_ assisted pyrolysis was investigated to promote cracking of organics [28]. Evidently, the yield of solid residue decreased from 84.8 wt.% to 81.8 wt.% after CO_2_ introduction, confirming the positive effect of CO_2_ on reduction of organics. However, the gas yield with different CO_2_ percentages exhibited a negligible difference.

#### 3.3.2. Solid Residues

The effect of pyrolysis temperature and atmosphere on the micro-morphologies of solid residues were further studied. As shown in Figure 4g-1,g-2, FeRS appeared as a thick fiber-like morphology, while a large number of floccus was distributed on the surface. It was speculated that the floccus might represent Fe filings or other metallic–oxides. FeRS and pyrolytic solid residues were magnetic, confirming the possibility of iron filings distribution. Interestingly, the solid residues obtained from pyrolysis of FeRS at 400, 500, and 600 °C exhibited abundant gills on the skeleton. However, further elevation of the pyrolytic temperature caused collapse of gills, but was accompanied by the appearance of pore structures, especially for the solid residue obtained from pyrolysis of FeRS at 900 °C. During pyrolysis, organics in FeRS were decomposed and pores were subsequently formed over residues. After the introduction of CO_2_, the density of gills increased on the skeleton, confirming that CO_2_ participated in the reduction of carbon skeleton.

The textual properties of FeRS and solid residues after pyrolysis were investigated via N_2_ adsorption–desorption tests. As shown in Figure 5, all samples exhibited type III adsorption isotherm with H3 hysteresis loops. The isotherm gradually increased at *P*/*P*_0_< 0.1 while rising sharply at *P*/*P*_0_> 0.8, indicating the coexistence of microporous and mesoporous structure [29]. The calculated BET surface area and detailed pore size distribution are summarized in Table 5. Clearly, the mesopores were dominant, while micropores only accounted for ~10% for all samples. FeRS possessed a surface area and pore volume of 13.04 m^2^/g and 0.19 cm^3^/g, respectively. After pyrolysis, the surface area and pore volume of solid residues elevated compared with FeRS. Organic decompositions released the original pores occupied by these organic matters. Moreover, gas formation and release during pyrolysis also contributed to the expansion of pore structures. In comparison, FeRS-400 exhibited a higher surface area and pore volume, ca. 52.15 m^2^/g and 0.056 cm^3^/g, respectively. With the elevation of pyrolytic temperature, the surface area and pore volume exhibited a decreasing tendency, originating from pore collapse at high temperature. Generally, CO_2_ assisted pyrolysis should be beneficial to pore formation by an etching effect on the carbon skeleton from the reaction with CO_2_ to generate CO. However, both specific surface area and pore volume of solid residues decreased after introduction of CO_2_. It can be speculated that CO_2_ provided an oxygen source for Fe oxidation to obtain more iron– oxides in the solid residues [28]. The transformation from zero-valent iron into iron–oxides was not favorable for pore formation. Therefore, an N_2_ atmosphere rather than CO_2_ was more suitable for improving the physical properties of solid residues due to the low content of carbon and high content of iron in FeRS. In short, the solid residues from FeRS pyrolysis possessed a porous structure with magnetic characteristics. In the future, some methods could be explored to modify its characteristics to produce Fe/C composite materials.

The XRD patterns of FeRS and solid residues from pyrolysis of FeRS at different conditions are presented in Figure 6 to identify the crystalline structures. These samples exhibited diffraction peaks at 45.0°, 64.5°, and 82.5°, corresponding to planes of Fe(110), Fe(200), and Fe(211), respectively. Moreover, a weak diffraction peak located at ~37.0° could be observed, corresponding to Fe_3_O_4_ [30]. Hence, zero-valent iron with different planes represented the dominant compositions. Compared with FeRS, solid residues displayed more intense diffraction peaks, suggesting a high degree of crystallinity after pyrolysis. The removal of organics in FeRS and high temperature calcination contributed to crystallization. There was no evident variation in the type of peaks but the intensity initially increased and then decreased with the elevation of pyrolytic temperature. FeRS-700 exhibited the highest intensity of characteristic peaks. Excessively high temperatures caused the breakage of skeletons, thus weakening crystallization. However, a CO_2_ atmosphere brought negligible effects upon phase compositions. In comparison, the diffraction peak corresponding to Fe_3_O_4_ slightly intensified under a CO_2_ atmosphere, confirming Fe oxidation.

#### 3.3.3. Pyrolytic Oil and Gas

In order to clearly study the effect of pyrolysis atmosphere on oil composition, all compounds monitored from GC–MS were identified into C_6_–C_10_, C_11_–C_15_, C_16_–C_20_, C_21_–C_25_, and >C_25_ compounds. As shown in Figure 7a, C_11_–C_15_, C_16_–C_20_, C_21_–C_25_, and >C_25_ compounds possessed the content of 30.2%, 30.0%, 27.5%, 12.2%, respectively, while no C_6_–C_10_ compounds were detected for pyrolysis of FeRS under an N_2_ atmosphere. However, C_6_–C_10_ compounds appeared in the pyrolytic oil and the proportion of C_11_–C_15_ also increased after 10% CO_2_ introduction, indicating the promotion of the cracking of C–C bonds from CO_2_. Unfortunately, the heavy fractions (C_21_–C_25_ and >C_25_ compounds) elevated when CO_2_ completely replaced N_2_, suggesting excessive CO_2_ was not conducive to the lighten oil compositions. Excessive amounts of CO_2_ might provide more C atoms and promote a polymerization reaction. GC was used to analyze the valuable components in pyrolytic gas. Clearly, H_2_ was the main component in pyrolytic gas, which attained the yield of 4.08 mL/g for pyrolysis of FeRS under N_2_ atmosphere. As the content of CO_2_ in atmosphere increasing, the H_2_ yield fluctuated. The H_2_ yield further elevated to 11.25 mL/g when 10% CO_2_ was introduced. On the whole, the H_2_ yield for all conditions under CO_2_ atmosphere surpassed that under N_2_ atmosphere. Besides, the yield of C_2_H_4_, C_2_H_6_, C_3_H_6_, and CH_4_ increased firstly but then decreased. 30% CO_2_ reached the highest yield with a total production of 17.76 mL/g. CO yield rose linearly with the content of CO_2_ due to the reaction between CO_2_ and carbon in solid residues, i.e., CO_2_ + C = 2CO. Further, water gas shift reaction, i.e., CO + H_2_O = CO_2_ + H_2_, contributed to H_2_ formation. Subsequently, parts of the oil compounds were cracked into micromolecular non-condensable gas compositions. Dramatically, CO_2_ introduction promoted LHV value of pyrolytic gas, and 30% CO_2_ introduction attained the highest LHV, ca. 3250 kJ/Nm^3^. Although the yield of gas was low during pyrolysis, CO_2_ introduction proved beneficial in attaining more pyrolytic gas with high LHV, thus replacing more auxiliary fuel for heat supply.

### 3.4. Effect of Heating Rate and Stirring

For industrial applications, the pyrolysis reactor includes a batch-type and continuous-type reactor, which exhibit different heating rates. Rotary kiln is widely applied in industry representing a rotation process that would promote heat and mass transfer during pyrolysis. Therefore, to investigate the effect of heating rates and stirring on pyrolysis of FeRS, rapid pyrolysis through direct placement into the reactor when the reactor temperature reached 500 °C (500-Q) and slow pyrolysis with different stirring rates (25 r/min and 50 r/min) in same heating rates (10 °C/min) and final temperature (500 °C), i.e., 500-25r and 500-50r, were conducted in the self-designed reactor. As shown in Figure 8a, the yields of solid residues were 84.8, 85.0, 86.6, and 87.4 wt.% for FeRS-500, 500-Q, 500-25r, and 500-50r, respectively. Heating rates exhibited negligible effects on products distribution, while stirring produced more solid residues but less oil production. However, higher yields of pyrolytic gas with stirring were observed, ca. 5.4 and 2.9 wt.%, while only 1.3 wt.% for FeRS-500. Therefore, stirring enhanced heat and mass transfer to promote cracking of oil compounds into pyrolytic gas. However, stirring might also promote coking during pyrolysis of FeRS, contributing to a slightly higher yield of solid residues. Furthermore, the proportions of C elements in FeRS-500, 500-Q, 500-25r, and 500-50r were 2.46, 4.28, 2.70, and 3.57 wt.%, respectively (Table 6). Higher heating rates were not favorable for complete degradation of organics in FeRS due to less reaction time, thus leaving more C contents in solid residues. Moreover, it may be caused by the excessive generation of carbonate compounds during the rapid heating process. The higher content of C in 500-25r and 500-50r further validated that enhanced coking by stirring. The mineral compositions in FeRS and solid residues are tabulated in Table 7. The proportions of total Fe possessed clear changes in FeRS, FeRS-500, 500-Q, 500-25r, and 500-50r, ca. 57.15, 74.91, 70.20, 69.09, and 63.65 wt.%. Clearly, elimination of organics from pyrolysis produced a high purity of iron in solid residues. However, higher heating rates and stirring brought more carbon containing compounds in solid residues. The crystal structures shown in Figure 8b demonstrate unchanged planes of Fe(110), Fe(200), and Fe(211), and small amounts of Fe_3_O_4_ under different conditions.

The nitrogen adsorption–desorption isotherms, pore size distribution, and textural properties of FeRS and obtained solid residues are presented in Figure 8c,d and Table 8. All samples showed type III adsorption isotherm with H3 hysteresis loop. Therefore, variation in heating rate and stirring also obtained co-existence of micropores and mesopores, while the proportion of micropores was 10%, approximately. The specific surface area of these samples decreased in the order: 500-50r > FeRS-500 > 500-25r > FeRS > 500-Q, ca. 44.73, 27.04, 13.57, 13.04, and 12.71 m^2^/g. Higher heating rates caused collapse of pores, thus obtaining a lower surface area and pore volume. To a certain extent, higher stirring rates were favorable for pore formation.

### 3.5. Resource Utilization of Pyrolysis Residues

As mentioned above, several pyrolytic parameters were investigated and results demonstrated that slow pyrolysis at 500 °C under an N_2_ atmosphere with a heating rate of 10 °C/min and stirring at 50 r/min should provide the optimal condition, which could achieve complete degradation of organics in FeRS and high quality of oil and solid residues. Due to the low content of oil in FeRS, solid residues after pyrolysis represent the dominant products after pyrolysis. To evaluate the application potential of solid residues, a series of characterization on solid residue (500-50r) were conducted. As presented in Table 9, the contents of non-metal elements (C/H/O/N/S) were extremely low and C/O possessed dominant proportions, ca. 3.57 and 4.73 wt.%, respectively. Interestingly, the density of FeRS and 500-5r was 0.47 and 0.56 cm^3^/g, respectively, which was lower than water. The practical photos of FeRS and solid residue (500-50r) in Figure 9a,b demonstrate loose powder, while FeRS floating on the water in Figure 9c also validates lower density. Moreover, Figure 9d displays the magnetic feature of the solid residue (500-50r). These observations provided guidance for recovery and further utilization with feasibility of separation of solid residues. The particle size distribution of FeRS and solid residue (500-50r) is presented in Figure 9e. Clearly, FeRS possessed a much broader range, which could be divided into two sections, ca. 0–500 μm and 500–3000 μm. After pyrolysis, the particle size concentration increased, i.e., 0–800 μm. If the solid residues require further treatment by sintering, granulation is essential to obtain a higher particle size to avoid being blown off in the sintering furnace. To evaluate the affinities for water of FeRS and solid residue (500-50r), contact angle tests were conducted. As shown in Figure 9f,g, the contact angles are both higher than 90^o^, implying hydrophobic characteristics.

The heavy metals in solid residues are also a critical index to evaluate the utilization potential. Figure 10a presents the content of heavy metals of FeRS and solid residues obtained in different conditions. Seven metals were detected, i.e., Fe, Cd, Cr, Cu, Ni, Pb, and Zn. Excluding Fe, six other metals could be recognized as heavy metals. Cr and Ni possessed relative higher concentration, ca. 5–15 g/kg. Next, the concentration of Cu and Zn ranged around 2 g/kg. Cd and Pb were scarce. Higher pyrolytic temperature reduced the content of Fe, indicating migration into oil and gas. CO_2_ introduction could reduce all the concentrations of metals, implying a release risk into oil and gas. The most concerned metal should be Cr as chromium–oxides exhibited high toxicity, especially for Cr^6+^. From XRF results in Table 9, except for iron–oxides, Al, Si, and Cr were the dominant minerals in 500-50r, ca. 7.89, 2.77, and 1.17 wt.%. Therefore, XPS measurements on FeRS and 500-50r were conducted for Cr 2p, as presented in Figure 10b. FeRS and 500-5r both exhibited co-existence of Cr(VI) and Cr(III). In comparison, the proportion of Cr(VI) increased after pyrolysis from 33.57% of FeRS to 52.42% of 500-5r. Hence, future utilization of solid residue should consider the detoxification and removal of the heavy metal Cr.

## 4. Conclusions

Iron-rich rolling sludge (FeRS) with hazardous characteristics significantly threaten human health and the ecological environment. Pyrolysis treatment can eliminate the organics in FeRS while obtaining valuable oil and gas. This paper investigated comprehensively the pyrolysis process of FeRS by TG-MS and fixed-bed reactor. FeRS contained less organics and large amounts of minerals with Fe as the dominant mineral. The activation energy for pyrolysis of FeRS was extremely low, ca. 5.44 kJ/mol. The effects of pyrolytic temperature, atmosphere, heating rate, and stirring on pyrolysis characteristics were conducted and analyzed. Commonly, the yield of solid residues maintained around 85 wt.%, with approximately 13 wt.% oil and 2 wt.% gas. CO_2_ introduction attained lighter oil compounds and more valuable gas due to promotion of cracking of oil compositions. The solid residues following pyrolysis exhibited porous properties with co-existence of micropores and mesopores. Quick pyrolysis would lead to the collapse of pores thus decreasing the surface area of solid residues. Zero-valent iron represented the dominant crystal structure of solid residues. Moreover, the particle sizes of solid residues were located at 0–500 μm with hydrophobic characteristics. The solid residues were suitable for sintering to further the utilization of resources. Moreover, due to the high content of zero-valent iron and its magnetic properties, it bears the potential to be applied in water pollution control and soil remediation. However, certain heavy metals, such as Cd, Cr, Cu, Ni, Pb, and Zn, were also trapped in the solid residues, which might cause secondary pollution upon their utilization. This paper lays a foundation for the harmless treatment of FeRS and the resource utilization of pyrolytic products.

## Figures and Tables

**Figure 1 ijerph-19-02152-f001:**
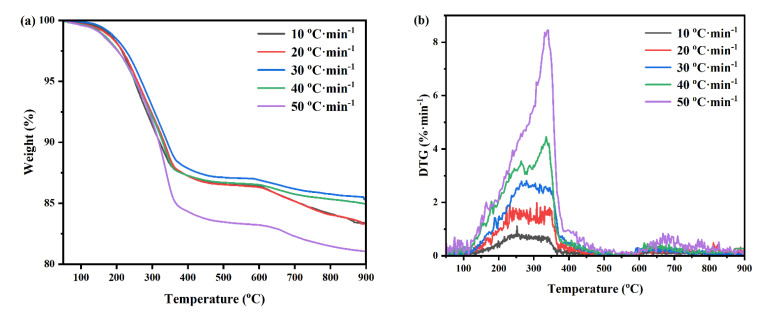
TG (**a**) and DTG (**b**) curves of FeRS under different heating rate.

**Figure 2 ijerph-19-02152-f002:**
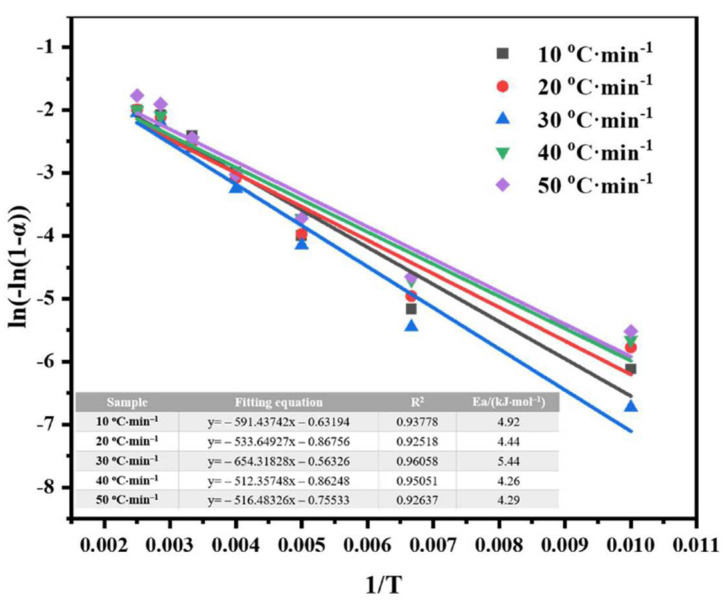
Kinetic analysis of FeRS pyrolysis.

**Figure 3 ijerph-19-02152-f003:**
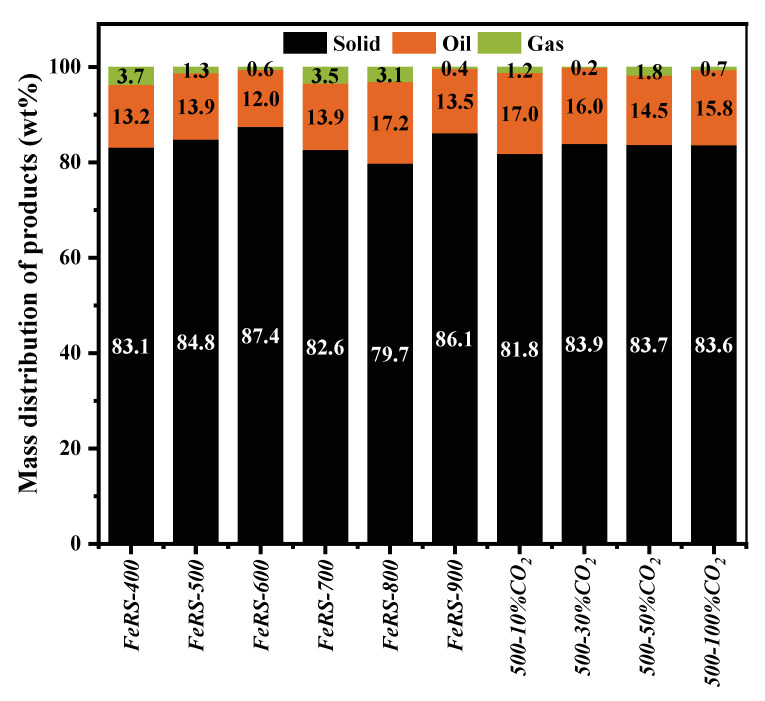
Pyrolytic products distribution under different temperature and atmosphere.

**Figure 4 ijerph-19-02152-f004:**
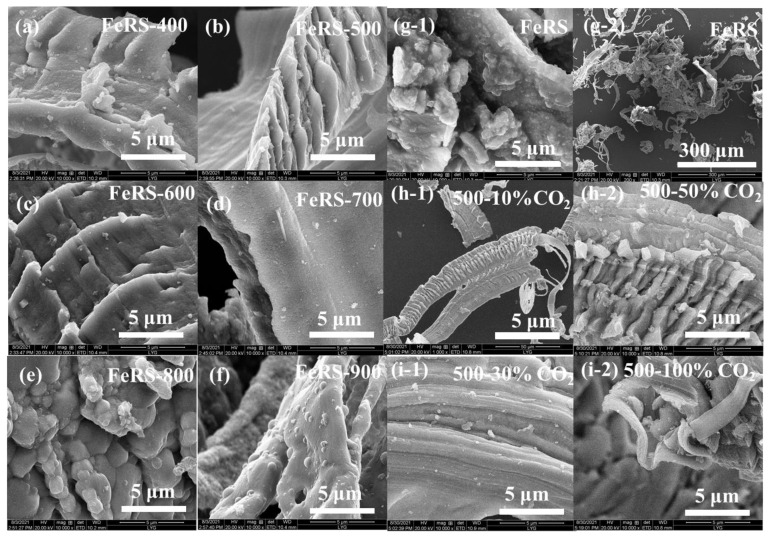
Morphology characterization of FeRS and solid residues from pyrolysis of FeRS at different conditions.

**Figure 5 ijerph-19-02152-f005:**
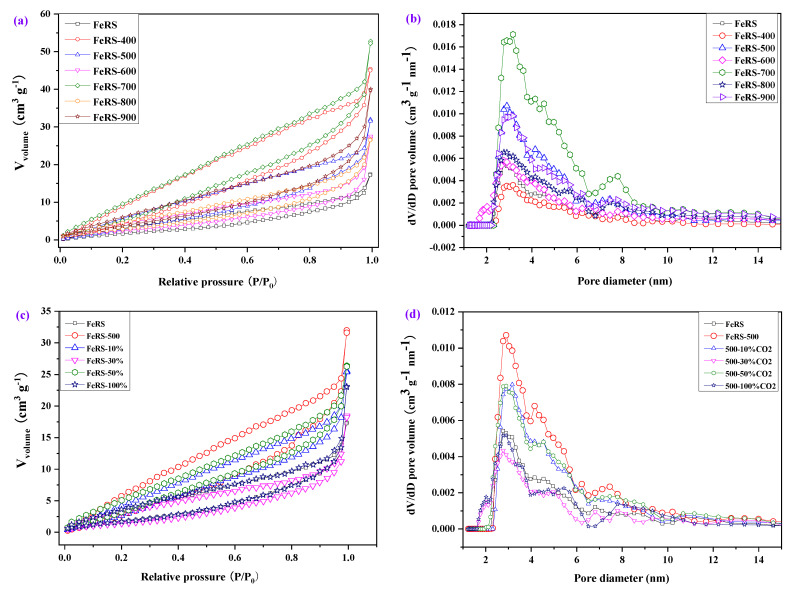
(**a**,**b**) Nitrogen adsorption–desorption isotherms and (**c**,**d**) pore size distribution of FeRS and solid residues from pyrolysis of FeRS at different conditions.

**Figure 6 ijerph-19-02152-f006:**
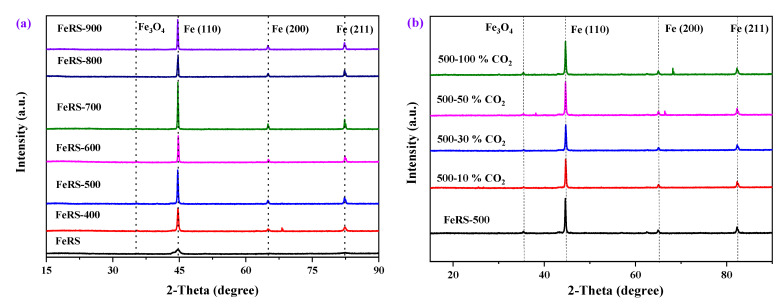
The XRD patterns of FeRS and solid residues from pyrolysis of FeRS at different conditions (temperature (**a**) and atmosphere (**b**)).

**Figure 7 ijerph-19-02152-f007:**
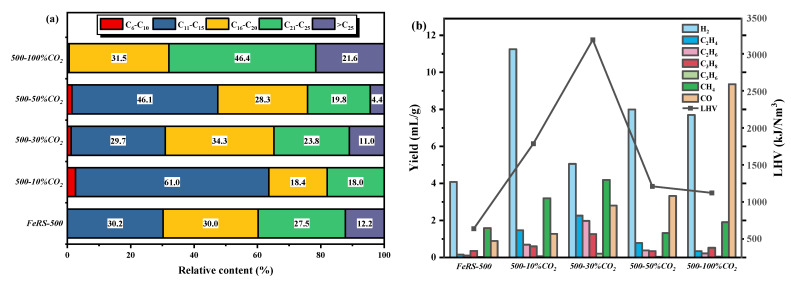
The relative content of components in pyrolytic oil (**a**) from GCMS results and yield of valuable components in pyrolytic gas (**b**) from GC results under different atmosphere.

**Figure 8 ijerph-19-02152-f008:**
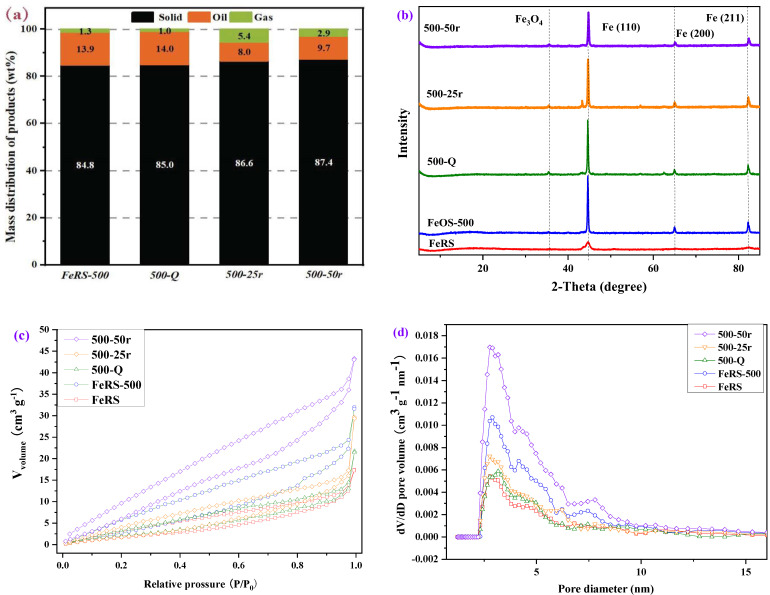
Pyrolytic products distribution under different heating rates and stirring (**a**); The XRD patterns of different samples in different heating rate and stirring (**b**); Nitrogen adsorption–desorption isotherms (**c**) and pore size distribution of FeRS and solid residues obtained at different heating rates and stirring (**d**).

**Figure 9 ijerph-19-02152-f009:**
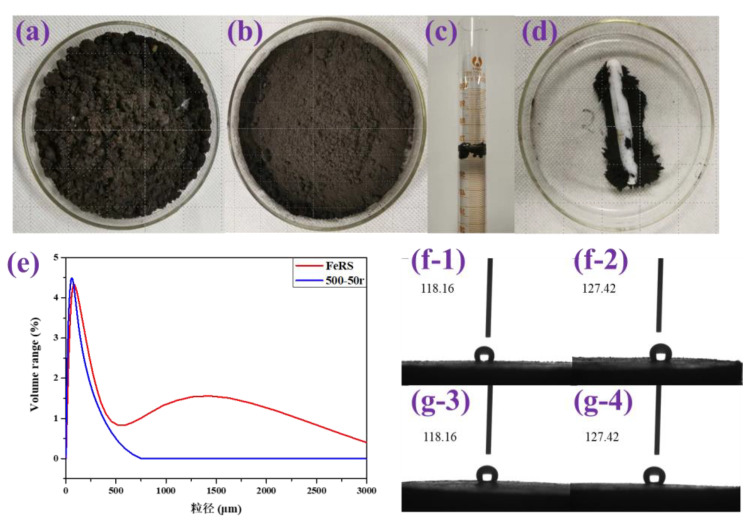
Photos of FeRS (**a**) and solid residue (500-50r) (**b**); the FeRS floating on the water (**c**); magnetic test of solid residue (500-50r) (**d**); particle size distribution of FeRS and 500-50r (**e**); contact angle test at different angles of FeRS (**f**) and 500-50r (**g**).

**Figure 10 ijerph-19-02152-f010:**
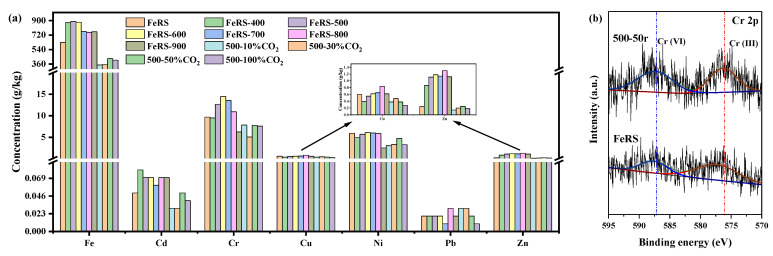
The content of heavy metals (ICP results) of FeRS and solid residues obtained in different conditions (**a**); Cr 2p XPS spectra of FeRS and solid residue (500-50r) (**b**).

**Table 1 ijerph-19-02152-t001:** Designation and experimental parameters.

Designation	Heating Rate (°C/min)	Ending Temperature (°C)	Stirring Rate (r/min)	CO_2_ Flow (mL/min)	N_2_ Flow (mL/min)
FeRS-400	10	400	0	0	100
FeRS-500	10	500	0	0	100
FeRS-600	10	600	0	0	100
FeRS-700	10	700	0	0	100
FeRS-800	10	800	0	0	100
FeRS-900	10	900	0	0	100
500-Q ^a^	--	500	0	0	100
500-25r	10	500	25	0	100
500-50r	10	500	50	0	100
500-10% CO_2_	10	500	0	10	90
500-30% CO_2_	10	500	0	30	70
500-50% CO_2_	10	500	0	50	50
500-100% CO_2_	10	500	0	100	0

^a^ Quick pyrolysis, the sample was placed into a reactor which reached the ending temperature.

**Table 2 ijerph-19-02152-t002:** The element and heating value analysis results of FeRS.

Sample	Element Analysis (%)					Heating Value Analysis (kJ/g)	
	C	H	O	N	S	HHV	LHV
FeRS	11.05	1.66	7.07	0.13	0.29	10.136	9.806

**Table 3 ijerph-19-02152-t003:** The mineral compositions of FeRS from XRF results.

Sample	Content (wt.%)										
	Na_2_O	Al_2_O_3_	SiO_2_	P_2_O_5_	CaO	Cr_2_O_3_	MnO	Fe_2_O_3_	NiO	CuO	MoO_3_
FeRS	0.06 ± 0.01	2.17 ± 0.2	1.48 ± 0.2	0.02 ± 0.003	0.05 ± 0.008	1.14 ± 0.1	0.43 ± 0.06	57.15 ± 4.1	0.37 ± 0.05	0.02 ± 0.004	0.38 ± 0.06

**Table 4 ijerph-19-02152-t004:** Kinetic fitting results of pyrolysis curves of FeRS.

Heating Rate	Fitted Equation	R^2^	Ea/(kJ·mol^–1^)
10 °C·min^–1^	y= −591.43742x − 0.63194	0.93778	4.92
20 °C·min^–1^	y= −533.64927x − 0.86756	0.92518	4.44
30 °C·min^–1^	y= −654.31828x − 0.56326	0.96058	5.44
40 °C·min^–1^	y= −512.35748x − 0.86248	0.95051	4.26
50 °C·min^–1^	y= −516.48326x − 0.75533	0.92637	4.29

**Table 5 ijerph-19-02152-t005:** Textural properties of FeRS and solid residues from pyrolysis of FeRS at different conditions.

Sample	Porosity Parameters				
	SSA (m^2^/g) ^a^	V_total_ (10^−1^ cm^3^/g) ^b^	V_micro_ (10^−1^ cm^3^/g) ^c^	V_meso_ (10^−1^ cm^3^/g)	V_micro_ /V_total_
FeRS	13.04	0.19	0.02	0.17	0.11
FeRS-400	52.15	0.56	0.05	0.51	0.09
FeRS-500	27.04	0.35	0.03	0.32	0.09
FeRS-600	12.93	0.27	0.03	0.24	0.11
FeRS-700	37.18	0.60	0.06	0.54	0.10
FeRS-800	15.40	0.31	0.03	0.28	0.10
FeRS-900	21.75	0.42	0.05	0.37	0.12
500-10% CO_2_	21.16	0.28	0.03	0.25	0.11
500-30% CO_2_	8.09	0.17	0.02	0.15	0.12
500-50% CO_2_	19.82	0.31	0.04	0.27	0.13
500-100% CO_2_	9.72	0.20	0.02	0.18	0.10

^a^: Multi-point BET specific surface area (SSA). ^b^: Total pore volume for pores with radius less than 14.72 nm at P/P_0_ = 0.99. ^c^: Total micropore (<2 nm) volume calculated using (Horvath–Kawazoe) H–K equation.

**Table 6 ijerph-19-02152-t006:** Element analysis of FeRS and residues from the pyrolysis of FeRS at different heating rates.

Sample	Element Analysis (%)				
	C	H	N	S	O
FeRS	11.05	1.66	0.13	0.29	7.07
FeRS-500	2.49	0.46	0.11	0.15	5.42
500-Q	4.28	0.51	0.15	0.19	6.10
500-25r	2.70	0.38	0.09	0.11	3.66
500-50r	3.57	0.45	0.11	0.12	4.73

**Table 7 ijerph-19-02152-t007:** Mineral composition analysis (XRF results) of FeRS and residues from the pyrolysis of FeRS at different heating rates.

Sample	Content (wt.%)										
	Al_2_O_3_	SiO_2_	P_2_O_5_	K_2_O	CaO	V_2_O_5_	Cr_2_O_3_	MnO	Fe_2_O_3_	NiO	MoO_3_
FeRS	2.17 ± 0.2	1.48 ± 0.2	0.02 ± 0.003	--	0.05 ± 0.008	0.36 ± 0.05	1.14 ± 0.1	0.43 ± 0.06	57.15 ± 4.1	0.37 ± 0.05	0.38 ± 0.06
FeRS-500	4.40 ± 0.8	0.93 ± 0.3	0.04 ± 0.007	--	0.06 ± 0.01	0.24 ± 0.07	1.27 ± 0.2	0.53 ± 0.1	74.91 ± 7.8	0.25 ± 0.06	0.32 ± 0.08
500-Q	3.79 ± 0.3	2.74 ± 0.1	0.05 ± 0.004	--	--	0.39 ± 0.04	1.50 ± 0.13	0.46 ± 0.04	70.20 ± 2.2	0.32 ± 0.03	0.41 ± 0.03
500-25r	4.05 ± 0.8	0.41 ± 0.1	0.41 ± 0.008	--	0.13 ± 0.01	0.58 ± 0.1	1.43 ± 0.3	0.51 ± 0.1	69.09 ± 8.8	0.40 ± 0.1	0.40 ± 0.1
500-50r	7.89 ± 1.3	2.77 ± 0.6	0.07 ± 0.01	0.03 ± 0.004	0.27 ± 0.02	0.38 ± 0.1	1.17 ± 0.3	0.42 ± 0.1	63.65 ± 6.0	0.35 ± 0.1	0.35 ± 0.1

**Table 8 ijerph-19-02152-t008:** Textural properties of FeRS and solid residues obtained at different heating rates and stirring.

Sample	Porosity Parameters				
	SSA (m^2^/g) ^a^	V_total_ (10^−1^ cm^3^/g) ^b^	V_micro_ (10^−1^ cm^3^/g) ^c^	V_meso_ (10^−1^ cm^3^/g)	V_micro_ /V_total_
FeRS	13.04	0.19	0.02	0.17	0.11
FeRS-500	27.04	0.35	0.03	0.32	0.09
500-Q	12.71	0.20	0.02	0.18	0.10
500-25r	13.57	0.24	0.02	0.22	0.08
500-50r	44.73	0.55	0.07	0.48	0.13

^a^: Multi-point BET specific surface area (SSA). ^b^: Total pore volume for pores with radius less than 14.72 nm at P/P_0_ = 0.99. ^c^: Total micropore (<2 nm) volume calculated using (Horvath–Kawazoe) H–K equation.

**Table 9 ijerph-19-02152-t009:** The element analysis, mineral compositions (XRF), and density of solid residue (500-50r).

Element Analysis (wt.%)					Mineral Compositions (wt.%)				Density (cm^3^/g)
C	H	O	N	S	Al_2_O_3_	SiO_2_	Cr_2_O_3_	Fe_2_O_3_	
3.57	0.45	4.73	0.11	0.12	7.89	2.77	1.17	63.65	0.56

## Data Availability

Not applicable.

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
