# Peer review of "Pyrolysis Behaviors and Residue Properties of Iron-Rich Rolling Sludge from Steel Smelting"

_ijerph, 2022, doi:10.3390/ijerph19042152_

Round 1

Reviewer 1 Report

The article investigated the pyrolysis process of iron-rich rolling sludge (FeOS). The products of the process at various conditions were characterized by several analytical techniques. I believe the novelty and originality of the paper is well explained in the introduction. The experimental section provides enough information, and the data are presented and discussed well. I recommend publishing the article after addressing the following minor comments:

Issue 1: Line 153: HHV and LHV stands for higher heating value and lower heating value. I know the other term is also correct but since you use this abbreviation I recommend to change.

Issue 2: Line 171 to 173: cite some reference to support these conclusions.

Issue 3: Line 183: Discuss more about the lower activation energy of FeOS compared to oily sludge. What does it imply?

Issue 4: 3.3.1: Does Figure 3 suggest there is no correlation between temperature and distribution products? There is no significant difference in product distribution when changing the temperature. Any idea why?

Issue 5: Line 202: “pyrolysis at 500 oC was enough to achieve completely elmination of organics in FeOS.” How this conclusion was made? Why not at 400 oC?

Issue 6: Line 218: How increasing temperature creates the gills?

Issue 7: Line 238: But pore volume reduced after heating to 0.05 cc/g. How heating increased BET but decreased pore volume?

Issue 8: Line 243: Please cite a reference for “It can be speculated that CO2 provided oxygen source for Fe oxidation to obtain more iron oxides in the solid residues”

Issue 9: Any idea why there is no correlation between BET SSA and temperature?

Reviewer 2 Report

Dear Authors I recommend a revision of your paper related to the application of analytical techniques. All methods, especially methods of elemental analysis, should be detailed described. Description of all methods should contain detailed data about equipment, provider, and sample preparation (there is no information for XRF, for example). Methods of elemental analysis (XRF, ICP MS) should contain data about calibration methods, reference materials (in they were used), standard deviation, limits of detection. For obtained data, it is necessary to add confidential intervals (tables 2,3,6,7). Please, check all abbreviations. I did not find FeOS abbreviation, it is not related to the name of the object (rolling sludge), maybe it should be changed to FeRS?Line 70 – decipherment should be ‘mass spectrometry’ Line 153-154 low calorific value – LCV, why it is LHV? Or it is low heating value. Please correct (for HHV also) Please add methods used to obtain data to the titles of tables and figures and somewhere to text (line 213 for example) Please add ‘a’ and ‘b’ to the title of figure 6. Table 9 – the only physical property in the table is density. Please change the title. Line 378 please delete ‘types’. There are all metals detected in the samples? Because some contents are very low, Information about limits of detection should be added to prove your data. You use for XRD equipment with Cu-anode, so there is high background level should be in patterns. There is no optimal condition for such types of objects, but if iron content is high, qualitative (and may be semi-quantitative analysis) is possible. Please add some comments about this. It seems like title of Tables 6 and 7 should be changed. Journal is dedicated to environmental and health, but only in line 40-43 it is noted related to investigated objects. I recommend to add some discussion about this field to ‘conclusion’ section.
